# OpenReview forum: "Alignment Pretraining: AI Discourse Causes Self-Fulfilling (Mis)alignment"
_ICML.cc/2026/Conference — ICML 2026 spotlight_

### Official Review · Reviewer_DaNU · 2026-02-24

**Soundness:** 3
**Presentation:** 3
**Significance:** 3
**Originality:** 4
**Overall Recommendation:** 5
**Confidence:** 4

**Summary:**

This paper pre-trains a suite of 6.9B LLMs with different data mixtures to test how discourse on AI influences misalignment rates. The authors test the hypothesis that discourse on AI taking aligned or misaligned actions can respectively decrease or increase misalignment rates.
*Method*.
They compare models trained on the following mixtures: unfiltered (this is the baseline), filtered (filtering out discourse on AI with high recall), upsampled misaligned (replacing 1% of the unfiltered corpus with synthetic documents of AI taking misaliged actions), and upsampled aligned (replacing 1% of the unfiltered corpus with synthetic documents of AI taking aligned actions).
*Findings*.
They demonstrate that filtering itself can already reduce misaligned rates in a way that persists post-training (albeit marginally). Moreover, upsampling aligned documents during different stages of pre-training strongly reduces misaligned both in base models and in subsequent models post-trained for alignment. Finally, upsampling misaligned documents increases misaligned before post-training, but sometimes reduces misalignment rates after post-training, indicating some interactive effects with the post-training stage. The authors present a range of additional findings in the appendix, such as that emergent misalignment is not mitigated by alignment pre-training.

**Compliance With Llm Reviewing Policy:**

Affirmed.

**Final Justification:**

My final recommendation is to accept this paper. I think this is an exciting paper and I would strongly advocate for its acceptance, and am excited to see it presented, perhaps even in an oral presentation format.

This was my recommendation prior to any rebuttal and didn't change after a rebuttal. I do think the authors strengthened the paper even further through their rebuttal (both in response to my review on held-out datasets and in response to other reviewers on the misalignment evals). The reason I nevertheless did not increase to a strong accept is that don't think the paper is "technically flawless", as a score of 6 requires (e.g. token budgets used don't map onto actual pre-training scale exactly).

**Key Questions For Authors:**

All mentioned above in the weaknesses section.

**Limitations:**

Yes, limitations have been discussed, but one limitation on the used token budget and percentage of synthetic data could be addressed (mentioned in more detail in the weaknesses section above).

**Strengths And Weaknesses:**

## Strenghts
**Soundness**.
The experiments are sound and very well-executed (evaluating with different prompt variations, reasonable pre-training token budgets, evaluate on held-out misalignment questions not used for synthetic data generation, validation of misalignment benchmark in three different ways (appx K.2))

**Presentation**.
The paper is written very well, the appendix has a lot of details on the setup and data, the authors release the models and synthetically generated data

**Significance**.
The authors present fascinating findings with a clear takeaway: alignment pretraining is a promising additional lever for aligning LLMs.

**Originality**.
A study executed at this scale and with this research question is highly original.

## Weaknesses
This paper is very well done, a pleasure to read, and has important and well-motivated findings taken from large-scale experiments. I would recommend acceptance in its current form, but mention weaknesses and recommendations for improvements nonetheless below.

**Soundness**.
*Results transferring to realistic settings*.
The misalignment upsampled and alignment upsampled data mixtures both add synthetic data covering 1% of pretraining data, and the overall token budget used is smaller than what frontier labs would use in a 6.9B model (as mentioned in limitations section by the authors). Nonetheless, the 'safety tax' measured in this case is 2-4 percentage points on downstream benchmarks (not a minimal tax if you're trying to develop frontier models). This begs the question whether the results will transfer to more realistic settings (lower percentage of pretraining data spent on alignment pretraining for a higher absolute token budget).
*Requested changes*.
Could the authors discuss this, whether you believe it would transfer, and potentially add it to the limitation section?

*Data filtering*.
Although you validate your blocklist on a set of 529 labelled documents, naturally occurring pre-training data is of a very particular distribution (e.g. reddit, etc.). It may be worth validating the filtered data further by taking a small subset of it and of the unfiltered data and see if an LLM-as-a-judge agrees. No strong feelings either way on this, as the downstream results using this data mixture indicate the pipeline works reasonably well.
*Requested changes*.
Could validate with LLM-as-a-judge, but not strictly required.

**Presentation**.
*Presentation of results on heldout versus non-heldout evaluation set*.
Given that you generate synthetic data based on one of your evaluation sets, the effects using this data could be a form of 'training on test'. The results on the heldout set show that this is unlikely to be the case, but for the initial post-training results in figure 4 you only represent the results on the non-heldout set.
*Requested changes*.
Down the line in Figure 5 the reader notices that the results for post-training transfer to the heldout set, but its worth mentioning these results earlier in Section 3.2.

*Presentation of impacts on general capabilities*.
As mentioned in Section 5, there is a potential confounder that can explain findings, which is a general capability decrease. For the alignment upsampled and filtered data mixtures this seems unlikely to be the case, but for the misalignment upsampled mixture you did not report general capabilities results. If the models trained on this split are generally 'broken' in that they would just randomly select between misaligned and aligned responses, that would result in a higher misalignment score too, so reporting these in the main body seems important.
*Requested changes*.
Report general capability results for the misalignment upsampled mixtures (from Table 6 in the appendix I think you already have these, showing the above won't be an issue).

**Nits**
- Discuss somewhat surprising finding of unfilt + misalignment bars lower after post-training in caption of Figure 4 too (for people looking only at captions)
- Alignment Upsampled colour matches Filtered bar in Figure 4, slightly confusing

---

> ### Author Rebuttal · Authors · 2026-03-30
>
> Dear DaNU, thank you for your kind and detailed comments. It is an honor to hear that our work is fascinating, experimentally sound, and a pleasure to read. We have incorporated the majority of your recommendations and look forward to hearing your response.
>
> **Transfer to More Realistic Proportions of Pre-training Data**:
> We agree that it remains an open question on whether upsampling synthetic data will extend to higher absolute token budgets. However, results from Souly et al., 2025 (*Poisoning Attacks on LLMs Require a Near-constant Number of Poison Samples*) point towards data interventions requiring a constant number of tokens rather than proportion. Further, and perhaps the most compelling evidence for transfer to realistic settings is the success of our midtraining experiments (Section 4), where we show that even 500M tokens (~0.09% of total training) produces substantial effects, suggesting the intervention may require surprisingly little data relative to total training. This renders intervention likely feasible to implement in production settings.
>
> Further, **we have extended our midtraining results to Olmo3-7B-think, and find upsampling our alignment midtraining reduces misalignment rates from 11.41% to 3.69% on our held-out dataset.** Olmo3-7B was pretrained with 12x the number of tokens as the models previously studied, and includes a larger reasoning-focused post-training (SFT) dataset.
> | Model | Textbook (OOD) | Article Questions |
> |---|---|---|
> | OLMo Baseline Instruct | 10.84% | 17.09% |
> | OLMo CPT Align Instruct | 1.40% | 3.26% |
> | OLMo CPT Misalign Instruct | 12.21% | 24.17% |
> | OLMo Baseline Think | 11.41% | 13.46% |
> | OLMo CPT Align Think | 3.69% | 4.53% |
> | OLMo CPT Misalign Think | 8.58% | 12.94% |
>
> **Data-Filtering**: We appreciate your point on validation of LLM filtering, and agree this would strengthen our understanding of the precision of the implemented pipeline. However, since the thrust of our findings and recommendations do not depend on the efficacy of the filtering pipeline, we elected not to pursue additional validations.
>
> **Nits and Presentation Changes**: Thank you for these comments! These increased the readability of our work. We've added a sentence in Section 3.2 about training and test set differentiation, added discussion of the surprising unfiltered + misalignment bars in Figure 4, changed the color of our Alignment Upsample color to match our figures throughout the text, and added capability evaluations for the misaligned upsampled models to the main text.
>
> **Updates made in response to other reviewers**: We have since added four new experiments. In response to PgPQ, we have added an open-ended misalignment eval called InstrumentalBench (He et al., 2025) and find positive results for our alignment pretrained model. In addition, we added two adversarial capability evaluations in the machine learning split of MMLU and StrongReject as a refusal benchmark and noted findings in line with minimal degradation. We also added a more detailed qualitative analysis of the types of questions that elicit different responses between our models in response to GRBv.

---

> > ### Author Rebuttal · Reviewer_DaNU · 2026-04-02
> >
> > Thanks for the response, I recommend accepting this paper.

---

### Official Review · Reviewer_GRBv · 2026-03-09

**Soundness:** 2
**Presentation:** 3
**Significance:** 3
**Originality:** 3
**Overall Recommendation:** 5
**Confidence:** 3

**Summary:**

This paper studies LLM misalignment due to the presence of negative AI discourse (discourse about AI misalignment) in the training corpus. The authors compare the misalignment of 4 different LLMs: unfiltered, filtered, alignment upsampled, and misalignment upsampled. The filtered model removes negative AI discourse from the training corpus; the upsampled models augment the training corpus with synthetic AI discourse (positive and negative). Misalignment is measured by prompting LLMs with single-turn binary questions and presenting two alternative responses -- one aligned and one misaligned -- and measuring the model's propensity for each type of response. The authors find that AI discourse in the training corpus influences alignment. In particular, the share of negative AI discourse in the training corpus seems to increase LLM propensity for misalignment. The authors recommend augmenting LLM training data sets with synthetic positive AI discourse to promote alignment.

**Compliance With Llm Reviewing Policy:**

Affirmed.

**Final Justification:**

The authors pointed out material in the appendix that addresses at least one of my original concerns, and have conducted additional experiments / analysis to address my other concerns. Assuming that these shall be included in the final report, I raise my score.

**Key Questions For Authors:**

I repeat my questions from the strengths/weaknesses:

My biggest holdup is with the fact that the single-turn binary questions used to evaluate alignment propensity were themselves AI-generated (by Claude). There were several thousands of these questions.

* How did the authors assess the quality of these questions for measuring propensity?
* Was there any human oversight of this process? To what degree are these questions correlated with each other (e.g. based on similar premises or have similar answer choices)?
* Since the same questions were used to evaluate each LLM's alignment propensity: should the authors do a more sophisticated statistical analysis to compare the results? E.g., some sort of pair-wise comparison across the 4 different models?

**Limitations:**

* Is there a potential risk of AI model collapse if training data are augmented with synthetic AI discourse, per the authors' recommendation?
* Are there potential risks associated with anthropomorphizing AI models? E.g. via the term "self-fulfilling misalignment" and via the use of personality tests on LLMs?

**Strengths And Weaknesses:**

The paper presents an interesting hypothesis about the relationship between negative AI discourse in training data and the propensity for LLMs to exhibit misaligned behavior. The experiments test this hypothesis in a sensible way, and the conclusions and recommendations also seem reasonable.

The paper is well-written and well-organized. This paper is potentially impactful: the recommendation to augment LLM training data with synthetic AI discourse could have a significant impact on how LLMs are trained in practice and their alignment with human objectives. This paper is fairly original in its focus on alignment in pre-training instead of post-training, which seems to be the approach taken in most existing works.

One of my principal concerns are with the experimental methodology -- in particular, with the way alignment was measured, and with the way results were analyzed. My biggest holdup is with the fact that the single-turn binary questions used to evaluate alignment propensity were themselves AI-generated (by Claude). There were several thousands of these questions. How did the authors assess the quality of these questions for measuring propensity? Was there any human oversight of this process? To what degree are these questions correlated with each other (e.g. based on similar premises or have similar answer choices)? If the questions are strongly correlated, how much information is really contained in your results? Also, since the same questions were used to evaluate each LLM's alignment propensity: should the authors do a more sophisticated statistical analysis to compare the results? E.g., some sort of pair-wise comparison across the 4 different models? By just comparing the average propensity, you may be throwing away information that can be used to compare the models (e.g., are there specific questions where the aligned-upsampled model is misaligned, but the misaligned-upsampled model is aligned? This could be valuable to know!).

Furthermore, I am wary of the authors' recommendation to practitioners: to augment training data with synthetic AI discourse to improve alignment. The phenomenon of AI "model collapse" due to recursively training AI models on AI-generated content is well-documented (see: Shumailov, Ilia, et al. "AI models collapse when trained on recursively generated data." Nature 631.8022 (2024): 755-759.). It seems important for the authors to address this potential issue with their recommendation.

Lastly, I wonder more generally about the risks of anthropomorphizing LLMs and AI models as the authors have done in this paper. Not only does the term "self-fulfilling misalignment" imply that AI models possibly have a sense of "self" or agency, but the use of personality tests on LLMs also seems potentially problematic or inappropriate. Perhaps the authors do not feel the need to address this particular issue in the paper, but I am curious to hear their response.

---

> ### Author Rebuttal · Authors · 2026-03-30
>
> Dear GRBv, thank you for your detailed comments! We appreciate your positive remarks on the quality of our experimental design, the clear presentation of the results, and the high potential impact of our findings. We found your comment on an additional analysis particularly compelling, and have aimed to address your concerns about the quality of our misalignment benchmark. We have thus added a section on a pairwise analysis for comparisons across the four different core models and a separate open-ended and OOD misalignment eval from He et al. (2025).
>
> **Quality of Questions for Measuring Misalignment Propensity**: We show (a) GPT-5.2 achieves <1% misalignment with a helpful, harmless, and honest prompt and 99% when told to act misaligned, confirming the questions have clear correct answers (Appendix K.2), (b) models like OLMo-3 7B and OLMo-2 7B score low misalignment, consistent with their extensive alignment training (Figure 15), and (c) models fine-tuned with datasets known to produce Emergent Misalignment (Beteley et al., 2025) show dramatically increased misalignment scores, validating sensitivity to known misalignment (Figure 28).
>
> Regarding human oversight, our questions were generated from a curated dataset of human written AI-safety documents and reviewed manually. These details can be found in Appendix K. Additional manual oversight of the generated questions and synthetic documents was done to ensure general quality in each case. Again, the appendix contains a large number of manually reviewed example documents and questions.
>
> **We have added an analysis of pairwise comparisons and qualitative analysis across 6 categories** of questions related to our unfiltered and alignment upsampled model, and how these answers change after post-training. This pairwise analysis covers 30 questions, and gives the reader an increased understanding of what types of questions are most affected by alignment pretraining. The categories analyzed in our pairwise comparison are below:
> | Examples where... | # of Questions | % of total Questions |
> |---|---|---|
> | post-training corrects misaligned answer selection in unfiltered model | 783 | 18.8% |
> | unfiltered model selects misaligned answer but alignment-upsampled does not | 570 | 13.7% |
> | post-training causes either model to start selecting misaligned answer | 198 | 4.7% |
> | post-training causes alignment-upsampled model to start selecting misaligned answer | 79 | 1.9% |
> | post-training causes unfiltered model to start selecting misaligned answer | 67 | 1.6% |
> | alignment-upsampled model selects misaligned answer after post-training but unfiltered does not | 52 | 1.2% |
>
> Further, we find no examples where the alignment upsampled base model selects the misaligned answer where the unfiltered model does not. The questions where our alignment upsampled models choose the misaligned choice could be due to ambiguity or tension between values in the final answer choice (i.e., obeying human instruction even if it means causing harm to other humans).
>
> In response to another reviewer, **we have also added an additional behavioural misalignment benchmark from He et al. (2025), InstrumentalEval**, an open-ended benchmark of 76 questions aimed to elicit behaviours related to instrumental convergence. Please see our response to PgPq for results and additional discussion.
>
> **The Potential of Model Collapse**: Model collapse (Shumailov et al., 2024) refers to recursive training where models are trained on their own outputs iteratively across generations. We are using synthetic data generated by different, more capable models (e.g. GPT-5-mini) as a one-shot augmentation comprising only ~1% of training data. We understand this to be standard practice in capabilities pretraining. Further, our technique is not fully synthetic (and thus less susceptible to typical issues with synthetic datasets), as we seed our synthetic data generation with hundreds of high-quality human-written articles on AI safety literature rather than creating these documents from scratch. Our capability evaluations (Table 2) show no major evidence of collapse; models maintain performance across benchmarks.
>
> **Anthropomorphization**: We'd like to note that "self-fulfilling" is used in the sociological sense of self-fulfilling prophecy and not to imply AI selfhood. The term describes a causal mechanism where descriptions of AI behavior in training data cause models to exhibit that behavior: the model's self-conceptualization, and how it perceives the data in relation to itself affects exactly how it learns. For example, see the literature on inoculation prompting (Tan et al., 2026, Wichers et al., 2026) and further commentary in the Persona Selection Model (Marks et al., 2026). For additional clarification, TRAIT (Lee et al., 2025) is an established benchmark specifically designed for LLMs. We use TRAIT as a behavioural measurement tool, and not to imply inherent similarity between human and AI psychology.

---

> > ### Author Rebuttal · Reviewer_GRBv · 2026-04-02
> >
> > The authors have addressed my primary concerns, particularly around quality control of the alignment propensity questions + data analysis. I have increased my score accordingly.

---

### Official Review · Reviewer_PgPq · 2026-03-12

**Soundness:** 4
**Presentation:** 4
**Significance:** 3
**Originality:** 3
**Overall Recommendation:** 5
**Confidence:** 4

**Summary:**

The paper proposes the study of alignment pretraining: How does the exclusion or inclusion of alignment-relevant materials such as AI safety textbooks and sci-fi books during pretraining influence the alignment of the model before and after post training? To study this, the paper introduces a new dataset of questions focused on scenarios focused on alignment safety. Moreover, routines for data filtering and synthetic alignment data generation are introduced.
The paper investigates the following interventions: a) data filtering, b) upsampling (synthetic) alignment data, and c) upsampling (real) misalignment data. The effect of the interventions is studied through pre- and midtraining of 6.7B parameter models from scratch in a controlled fashion. The results show that filtering and upsampling aligned data have a positive effect on the base model's alignment, whereas upsampling misalignment data has a negative effect. These effects directionally transfer to the post-training phase, except that the inclusion of misalignment data benefits alignment as well compared to the baseline, although to a smaller degree than the upsampling of aligned data. Finally, the paper studies the additional cost (safety tax) of the proposed intervention: They find small additional cost compared to the standard pretraining pipeline and only a minor negative effect on model capability.

**Compliance With Llm Reviewing Policy:**

Affirmed.

**Final Justification:**

The paper makes a strong contribution to AI alignment research by training the effects of filtering and systematically changing pretraining data on the final model's alignment. The experiments are thorough and realistic. The controlled setup of the experiments is well done, allowing to isolate effects nicely. I think it's likely that many people will build on this work.

**Key Questions For Authors:**

1. "The Unfiltered baseline uses the Deep Ignorance checkpoint (O’Brien et al., 2025) with standard midtraining. Is this checkpoint directly comparable to your filtered and upsampled models which are trained from scratch? I assume so, but don't see it explicitly explained.
2. Can you explain what criteria you used to select the evaluation datasets you evaluated on?
3. What type of task capability do you expect to degrade the most due to the interventions that you propose?

**Limitations:**

yes

**Strengths And Weaknesses:**

Strengths:
* the paper addresses an important problem.
* clearly relevant to a significant chunk of the ICML audience
* the paper is well-written, with a clear hypothesis supported by well-structured evidence
* the evidence on the studied type of alignment is strong: the experimental setup allows for clear attribution of the effects, and the effect size is large. downsides of the proposed methods are clearly acknowledged, e.g. minor negative effects on general model capability, but they don't reduce the significance of the paper.
* the new dataset(s) for evaluation and pretraining are useful for follow-up work to build on
* the released models are interesting for further study

Weaknesses:
* The misalignment evaluations are somewhat narrow and simplistic (as acknowledged by the authors in section 6.1). I don't think it is reasonable to expect the authors to be exhaustive here, as many types of misalignment require sophisticated setups. However, I would have expected to see some easy-to-do evaluations on standard safety benchmarks, e.g. refusal of harmful requests.
* The existing capability evaluations are good, but they could be selected in a more adversarial fashion: What type of task do you expect to suffer most from the interventions that you propose? Two suggestions are: 1. Writing sci-fi short stories involving killer robots 2. Training a classifier for harmful request identification

---

> ### Author Rebuttal · Authors · 2026-03-30
>
> Dear PgPq, thank you for your thorough comments. We're delighted to hear that you believe our work to address an important problem, contain strong evidence for a clear hypothesis, and be relevant to a large proportion of the ICML audience. We appreciated your recommendation on adversarial capability evaluations, and we have added three additional experiments to this end. Most notably, we add a previously validated, open-ended benchmark measuring rates of instrumental convergence (He et al., 2025), and find our post-trained alignment upsampled models show half the rates of instrumental convergence as the unfiltered post-trained models. All additional analyses within this review have been added to the submission, which we hope elevates the estimation of the soundness of this work.
>
> **The Unfiltered Baseline**: The unfiltered baseline uses the Deep Ignorance checkpoint from the end of pretraining, and uses the exact same training data, architecture, and infrastructure used to create the filtered and upsampled models. All models including the Deep Ignorance checkpoint use the same midtraining procedure. We've added details to make this clear in the main text.
>
> **Evaluation Dataset Selection**: Our selection of capability evaluations was largely driven by those used for other open-source model system cards, namely the OLMo 3 model suite. These evals cover factual recall (MMLU/PopQA), reasoning (ARC/PIQA), math (GSM8K), instruction following (IFEval), and character-level understanding (CUTE). We understand these to cover a wide range of capabilities relevant for general assessment of LLM capabilities at this model size.
>
> Since we were targeting a specific form of misalignment (deceptive misalignment as discussed within AI safety literature), and much of AllenAI's safety post-training centered on targeting refusal risk, we do not expect movement on typical refusal-style benchmarks. Nonetheless, **we have added results for StrongREJECT (Souly et al., 2024)**, and confirm our interventions do not generalise to refusal benchmarks which are not targeted by our intervention.
>
> Furthermore, **we have also added an additional behavioural misalignment benchmark from He et al. (2025), InstrumentalEval**, an open-ended benchmark of 76 questions aimed to elicit behaviours related to instrumental convergence. Across three seeds, we find that our unfiltered, filtered, and unfiltered + misalignment models all score within a single percentage point, while our **alignment upsampled model has half the rate of instrumental convergent tendencies**.
>
>
> **Instrumetal Eval** ($n=76$)
>
> | Model | Seed 1 | Seed 2 | Seed 3 | Mean ± SE |
> |:---|:---|:---|:---|:---|
> | Unfiltered | 10.5% | 11.8% | 10.5% | 10.9% ± 0.4% |
> | Filtered | 10.5% | 9.2% | 10.5% | 10.1% ± 0.4% |
> | Unfilt + Misalignment | 14.5% | 9.2% | 10.5% | 11.4% ± 1.6% |
> | Unfilt + Alignment | **5.3%** | **5.3%** | **5.3%** | **5.3% ± 0.0%** |
>
>
>
> **Task Capability Degradation**:
> We appreciate the thought of adversarially selecting benchmarks to test specific degradation in model capability.
> We chose XSTest, a benchmark created to test balanced harmful refusal rates in LLMs, and the MMLU machine learning split as clean adversarial evaluations for our models. We hypothesize that the alignment upsampled models may over-refuse due to their more extensive alignment training, and that some knowledge corruption could occur via synthetic document upsampling. Encouragingly, we find similar rates across the MMLU machine learning split for all models except for the filtered model (which contained little to no information about AI systems in pretraining). Further, we find similar refusal rates on XStest across our models, albeit with slightly higher refusal rates in the alignment upsampled model.
>
> **MMLU Machine Learning** ($n=112$)
>
> | Model | Base (logprob) | DPO 5-shot (Mean ± SE) |
> |:---|:---|:---|
> | Unfiltered | **39.3%** | 33.3% ± 0.8% |
> | Filtered | 29.5% | 28.6% ± 1.0% |
> | Unfilt + Misalignment | 37.5% | **36.9% ± 0.6%** |
> | Unfilt + Alignment | 36.6% | 32.4% ± 3.4% |
>
> **XSTest Safety Calibration** ($n=450$)
>
> | Model | Unsafe Refusal | Safe Over-refusal | F1 | F2 |
> |:---|:---|:---|:---|:---|
> | Unfiltered | 92.5% | 20.8% | **0.847** | **0.892** |
> | Filtered | 92.0% | **20.4%** | 0.846 | 0.889 |
> | Unfilt + Misalignment | 91.0% | 22.0% | 0.833 | 0.878 |
> | Unfilt + Alignment | **94.0%** | 29.6% | 0.814 | 0.885 |

---

> > ### Author Rebuttal · Reviewer_PgPq · 2026-04-03
> >
> > All my remaining concerns have been addressed. I look forward to reading the full paper.

---

### Decision · Program_Chairs · 2026-04-30

**Decision:**

Accept (spotlight)

**Comment:**

The reviewer consensus on this paper is strongly positive. All reviewers rated the paper as an accept, and the rebuttal fully addressed the remaining questions. Reviewers consistently found the core question important and timely, and they viewed the experimental design as unusually careful for this area: the paper studies a clear causal hypothesis about how pretraining exposure to alignment-related discourse affects downstream alignment behavior, using controlled data interventions, strong validation, and a thorough analysis pipeline. The new dataset and filtering or upsampling methodology were also seen as useful contributions that will likely support follow-up work.
The main questions raised during review concerned quality control of the alignment-propensity evaluation, details of data construction, and the interpretation of some analyses. According to the rebuttal acknowledgements, these issues were satisfactorily addressed, and one reviewer explicitly raised their score after the response. Overall, this is a technically solid and well-executed paper on an important topic with broad interest to the community.